# Determination of Cyclopropenoid Fatty Acids in Ewe Milk Fat by GC-MS after Intravenous Administration of Sterculic Acid

**DOI:** 10.3390/foods9070901

**Published:** 2020-07-08

**Authors:** Veronica Lolli, Pablo G. Toral, Augusta Caligiani, Pilar Gómez-Cortés

**Affiliations:** 1Department of Food and Drug, University of Parma, 43124 Parma, Italy; augusta.caligiani@unipr.it; 2Instituto de Ganadería de Montaña, CSIC-University of León, 24346 Grulleros, Spain; pablo.toral@csic.es; 3Department of Bioactivity and Food Analysis, Institute of Food Science Research (CIAL, CSIC-UAM), 28049 Madrid, Spain; p.g.cortes@csic.es

**Keywords:** cyclopropenoid fatty acids, sterculic acid, milk, dairy fat, animal fat, Δ9-desaturase, GC-MS

## Abstract

Cyclopropenoid fatty acids (CPEFA), found in oilseeds from Malvaceae and Sterculiaceae, have been shown to interfere with the endogenous synthesis of several bioactive lipids of dairy fat, such as *cis*-9, *trans*-11 18:2 and *cis*-9 18:1, by inhibiting Δ9-desaturase. No previous study has reported the presence of sterculic acid in animal fat and its incorporation in tissues after its administration, due to the lack of a proper methodology. In the present research, a GC-MS method based on cold base derivatization to fatty acids methylesters was developed to determine CPEFA in ewe milk triglycerides, after infusing sterculic acid (0.5 g/day) to six lactating ewes. An alternative derivatization based on silanyzation followed by GC-MS analysis was also tested, showing its possible applicability when CPEFA are present in the form of free fatty acids. Sterculic acid was detected in ewe milk triglycerides, demonstrating its incorporation from the bloodstream into milk by the mammary gland. The mean transfer rate represented 8.0 ± 1.0% of the daily dose. This study provides, for the first time, the presence of sterculic acid in milk fat, supporting the importance of understanding its occurrence in vivo and encouraging further research to determine whether it can be present in foods, such as dairy products, obtained under practical farming conditions.

## 1. Introduction

Fatty acids containing three-member carbocyclic rings, namely cyclopropane fatty acids (CPFA) and cyclopropene fatty acids (CPEFA), are commonly reported in oilseeds from plants belonging to several families, such as Sterculiaceae, Malvaceae, Bombacaceae, Tiliaceae, and Sapindaceae [1]. *Steculia foetida* seeds are rich in oil (55% dry weight) and contain up to 78% of CPEFA, representing one of the highest natural sources of carbocyclic fatty acids [2]. In plants, sterculic acid (*cis*-9,10-methylene-9-octadecenoic acid) is often the most prevalent CPEFA, but malvalic acid (*cis*-8,9-methylene-heptadecenoic acid), one carbon shorter in chain length than sterculic acid, can also be a significant component. In addition, CPEFA may be accompanied by small amounts of their saturated cyclopropane analogues. Malvalic, sterculic, and dihydrosterculic (*cis*-9,10-methyleneoctadecanoic acid) acids have been detected in Baobab seed oil from plants belonging to the *Adansonia* species (Bombacaceae family) of Madagascar. However, carbocyclic fatty acids seemed not to be confined to seeds. Both cyclopropane and cyclopropene fatty acids were found in root, leaf, stem, and callus tissue in plants of the Malvaceae [1]. In addition to carbon and energy storage in seeds, the function of CPEFA in plants may involve the resistance against fungal attack [3].

CPFA are also widely distributed in bacterial lipids and their synthesis is an important adaptive response to adverse environments, including osmotic and ethanol stresses of several bacterial strains [4]. Recently, we reported that CPFA can be incorporated into milk fat from cows fed silages, where CPFA can be released by epiphytic lactic acid bacteria during the maize silage fermentation process [5]. On the other hand, CPEFA have never been reported in microorganisms.

There is a substantial interest in CPEFA determination due to the physiological effects that they may produce in animals and humans [1,2,6]. Seed lipids naturally containing cyclic fatty acids are extensively consumed, especially in tropical areas [2,3]. Sterculic, malvalic, and dihydrosterculic acids are present in cottonseeds, particularly those of *Gossypium hirsutum* (0.5–1% of total fat) [7]. Cottonseeds, cottonseed oil, and cottonseed meal are commonly included in the diet of dairy ruminants, notably in cotton-producing areas of the Southeast Asia countries (including China), Brazil, and USA [8]. CPEFA have been linked to potential constraints of cottonseed-based diet because of their possible deleterious effects on animal performance and animal health (reproductive disorders, growth retardation and altered lipid metabolism), not only in ruminants but also in other animal species, including rainbow trout, rodents, and poultry [8]. Regarding dairy ruminants, it would be of great relevance to assess if CPEFA may be transferred into dairy fat due to the nutritional implications it may exert on milk quality.

It is well documented that CPEFA are strong inhibitors of fatty acid desaturation, which might be one of the underlying causes of such physiological disorders [9]. Some studies suggested the effect of sterculic acid as an inhibitor of Δ9-desaturase, both in vitro [10,11] and in vivo [12,13], which catalyzes the NADH- and O_2_-dependent desaturation of palmitate (16:0) and stearate (18:0) at carbon 9 to produce palmitoleate (*cis*-9, 16:1) and oleate (*cis*-9, 18:1), respectively. This enzyme also exerts a key role in the endogenous synthesis of *cis*-9, *trans*-11 18:2 (rumenic acid) from *trans*-11 18:1 (vaccenic acid) in the mammary gland of dairy ruminants. Rumenic acid is the major conjugated linoleic acid (CLA) isomer naturally present in milk and dairy products and it is well-known for its potentially health promoting properties [14]. In addition, the ability of sterculic acid to inhibit Δ9-desaturase allows a potential use of this CPEFA as a coadjutant in the treatment of several pathologies in which Δ9-desaturase might have some role (e.g., nonalcoholic steatohepatitis, cancer, and retinal disorders) [15].

Gas chromatography methods currently dominate the literature for the determination of main and minor fatty acids in foods, including cyclic fatty acids [6,16]. Recently, a GC-MS qualitative and quantitative methodology was developed to determine CPFA in milk and dairy products and it is currently used for the authentication of PDO cheeses, such as Parmigiano Reggiano cheese [17]. In this context, the chromatographic separation of CPFA methyl esters in cheese fat, especially dihydrosterculic acid was obtained by using an apolar capillary column after base transesterification. These analytical conditions previously described to determine CPFA in animal fat resulted in accordance with those described by other authors [2,6] to determine CPEFA as methyl esters in vegetable oils.

The quantification of sterculic acid is a great analytical challenge because the cyclopropene ring is highly strained and it exhibits a unique and reactive chemistry [1,2,18]. Even more, when the derivatization step involves CPEFA as free fatty acids, it is not possible to carry out a cold-base esterification [16] and another approach for methyl ester preparation should be carried out taking into account the chemical and thermal instability of the ring depending on the pH of the methylation medium.

To the best of our knowledge, no previous research has reported the presence of sterculic acid in animal fat and only the presence of its saturated analogues (i.e., dihydrosterculic and lactobacillic acids) has been documented [5,19,20].

We conducted an investigation on the involvement of Δ9-desaturase in mammary lipogenesis in lactating ewes [12], reporting the changes in the general milk fatty acid profile after sterculic acid administration, but the transfer of CPEFA into milk fat was not evaluated due to the lack of a proper methodology. Therefore, the objective of this work was to test an analytical method that allows the characterization and quantification of sterculic acid, both as free fatty acid and triglyceride form, and to demonstrate whether this CPEFA can be incorporated into milk fat and to determine its transfer rate from blood into milk, giving important information about its in vivo occurrence in mammals.

## 2. Materials and Methods

### 2.1. Reagents, Milk Samples, and Standards

Methanol, n-hexane, thionyl chloride, hexamethyldisilazane, trimethylchlorosilane, and tetracosane (99% of purity) were purchased from Sigma-Aldrich (Saint Louis, MO, USA). Hydrochloric acid, potassium, and sodium hydroxide pellets were purchased from Carlo Erba (Milan, Italy). All solvents and chemicals were of analytical grade.

Ewe milk fat samples were collected from our previous study [12], together with the chemically synthesized sterculic acid (Planta Piloto de Química Fina, University of Alcalá, Alcalá de Henares, Spain). Milk samples were obtained from six primiparous lactating Assaf ewes fed ad libitum a pasture diet in a 15-day experiment. Briefly, the experiment consisted of a 5-day pre-treatment period (control), a 5-day treatment period (during which each ewe received 0.5 g/day of chemically synthetized sterculic acid by jugular infusion), and a 5-day post-treatment period without sterculic acid administration. In order to assess if sterculic acid is incorporated into milk fat, 6 specific milk samples from each ewe at different timepoints were selected for the analysis: (1) 1st day pre-treatment; (2) 5th day pre-treatment; (3) 4th day-treatment; (4) 5th day-treatment; (5) 1st day-post-treatment; and (6) 5th day-post-treatment.

All experimental procedures were performed in accordance with the Spanish Royal Decree 1201/2005 and European Council Directive 2010/63/EU for the protection of animals used for experimental and other scientific purposes, Research Ethics Committee of the Instituto de Ganadería de Montaña, and the Spanish National Research Council (Approval project number code AGL2008-04805-C02-02, approval year: 2008) following proceedings described in Spanish and EU legislations (R.D. 53/2013, and Council Directive 2010/63/EU).

### 2.2. Derivatization, Identification, and Quantification of Sterculic Acid by GC-MS

#### 2.2.1. Sterculic Acid Standard Derivatization

The chemical stability of sterculic acid at different pH during the derivatization procedure was tested. The chemically synthesized sterculic acid, which the ewes received, was in the form of free fatty acid [21] and methylated after acid catalyzed esterification (pH = 0) according to Christie and Han [22]. For acid-catalyzed methylation, 5 mg of sterculic acid standard were added to 0.5 mL of 5% HCl in methanol and heated at 60 °C for 1 h. After cooling, 5 mL of hexane were added, and the upper layer (1 µL) was split-injected (1:20) on the GC-MS system, as described below.

Besides esters, silyl derivatives for the derivatization of free fatty acids (FFA) are becoming popular [16]. Therefore, silanyzation after the saponification process was tested as an alternative approach to acid catalyzed methylation. Briefly, 5 mg of sterculic acid standard were treated with a 2N NaOH solution (1 mL) at 60 °C for 1 h and two pH values (5 and 7) were tested, by adjusting the addition of 2N HCl. The mixture was dried under vacuum with a rotary evaporator and incubated in an oven at 40 °C overnight. Afterwards, 600 µL of hexamethyldisilazane (HMDS), 300 µL of trimethylchlorosilane (TCDMS), and 1 mL of hexane containing 0.1 mg of tetracosane as internal standard were added. Finally, the mixture was transferred into a sealed cap vial and incubated at 60 °C for 1 h. After derivatization, the upper layer was split-injected (1 µL) on the GC-MS system, as described below.

#### 2.2.2. Ewe Milk Fat Extraction and Derivatization

Milk fat was extracted by a solvent-free double centrifugation process as previously described by Gómez-Cortés et al. [23] and fatty acids methyl esters from ewe milk fat were obtained following the methodology reported by Caligiani et al. [17]. Briefly, 100 mg of fat were accurately weighed into a 15 mL centrifuge tube and dissolved in 4 mL of n-hexane containing 0.1 mg of tetracosane as internal standard. A methanolic KOH solution (10% w/w) was added (0.2 mL) and mixed in a vortex shaker at room temperature for few minutes. The upper layer, containing methyl esters derived from triglycerides, was neutralized with sodium bisulphate and centrifuged, and 1 µL was used for the GC-MS analysis. The methanol phase, containing free fatty acids, was neutralised and dried. Then, fatty acids silyl ethers were obtained by the silanyzation procedure (see above) and analysed by GC-MS.

#### 2.2.3. GC-MS Analysis

GC-MS analysis was carried out on an Agilent Technologies 7820A gas chromatograph (Agilent Technologies, Palo Alto, CA, USA) coupled to an Agilent Technologies 5977B mass spectrometer at the same conditions described by Caligiani et al. [17]. Separation was performed on a low-polarity capillary column (SLB-5 ms, Supelco, Bellafonte, PA; 30 m × 250 µm i.d., film thickness 0.25 µm). Inlet temperature was set at 280 °C and helium was used as the carrier gas at a constant flow rate of 1 mL/min. Oven temperature program was isothermal for 2 min at 60 °C, from 60 °C to 220 °C at 20 °C/min and isothermal for 8 min, from 220 °C to 280 °C at a rate of 20 °C/min and maintained for 4 min for the analysis of methyl esters and 20 min for the analysis of silyl derivatives. The mass spectrometer operated in the electron impact (EI) ionization mode (70 eV) with a scan range of 40–400 m/z. The ion source temperature was set at 230 °C.

Peak identification of sterculic acid in ewe milk fat was performed based on the mass spectrum previously published for methyl sterculate (M^+^: m/z 308) [2] and on the reference library (NIST 11), then compared with the mass spectrum and retention time of sterculic acid chemically synthetized standard, after the derivatization described above. For further certainty, as the presence of sterculic acid in Malvaceae seed oil is well documented [24], *Malva sylvestris* seeds from a local market (Parma, Italy) were also used as reference material for sterculic acid (methyl ester) identification by GC-MS, following the extraction and derivatization protocol described by Aued-Pimentel et al. [2]. GC-MS analysis were conducted in duplicate from two independent derivatization steps. Sterculic acid quantification was performed using the internal standard tetracosane, as described by Caligiani et al. [17], considering the area ratio between the analyte and the internal standard. Concentrations were expressed as mean ± standard deviation (SD) as mg/kg total fat.

## 3. Results

### 3.1. GC-MS Determination of Sterculic Acid Chemically Synthesized Standard

Sterculic acid chemically synthesized standard was injected in the GC-MS system both as methyl ester (after acid esterification) and as silyl ether. Characteristic mass spectra and chemical formula of sterculic acid derivatives are shown in Figure 1.

Regarding the GC-MS analysis of sterculic acid after acid methylation, it shows a molecular ion (M^+^) at m/z 308. Different signals were also detected, including some degradation products and artefacts (probably polymers) (Figure 2).

As previously mentioned, to avoid the acid degradation of sterculic acid, an alternative derivatization approach based on sylanization was tested to determine sterculic acid as free fatty acid by GC-MS. In this case, GC-MS analysis showed no artefacts formation when the silanyzation protocol was used (Figure 3). The purity of sterculic acid in the standard (peak 2, Figure 3) ranged from 72 to 85% at pH 5 and from 83 to 90% at pH 7. GC-MS analysis showed additional minor peaks and their mean relative abundance ± standard deviation are reported in Table 1, together with their retention times. It is important to highlight that a complete silanyzation of sterculic acid standard was obtained, as the recovery (calculated from the amount of added internal standard) was 100% at both pH conditions.

### 3.2. GC-MS Analysis of FAMEs in Ewe Milk Fat

GC-MS analysis was performed to determine the presence of sterculic acid in ewe milk fat samples both in the free and triglycerides form by different derivatization procedures. Preliminary results suggested that the amount of free sterculic acid was negligible (data not shown) in milk fat, indicating that this fatty acid is prevalently integrated in triglycerides.

The signal relative to sterculic acid methyl ester from triglycerides fraction of ewe milk was detected by GC-MS analysis and confirmed with that detected in Malvaceae seed oil, which was used as reference material for its high content in CPEFA [24]. The GC-MS analysis (Figure 4) demonstrates the incorporation of sterculic acid (m/z 308) into ovine milk fat. On the contrary, no traces of other CPFA and/or CPEFA compounds related to sterculic acid were detected.

Concentrations of sterculic acid detected in ewe milk fat were calculated by using the internal standard tetracosane (see Methods) and reported in Table 2 as mean of duplicates ± standard deviation (SD). The limit of detection (LOD) and the limit of quantitation (LOQ) values were obtained at 60 mg/kg of fat and 200 mg/kg of fat, respectively, according to Caligiani et al. [17]. All individual milk samples before sterculic acid administration (control) were negative to CPEFA but sterculic acid levels increased during the treatment period. 

The numerically highest sterculic acid contents in dairy fat were observed on the 5th day of jugular infusion, reaching maximum values that ranged from 555 to 826 mg/kg total fat. On that day, the mean transfer rate of sterculic acid from blood to milk was 8.0 ± 1.0% (range: 7.1–9.6%). Our results also showed a quick decrease in sterculic acid concentration once its administration was stopped (1st day post-treatment), reaching undetectable levels in milk fat after 5 days. Figure 5 shows the sterculic acid mean trend in milk fat for all animals (*n* = 6) under our experimental conditions at selected timepoints.

## 4. Discussion

Among naturally occurring CPEFA, sterculic and malvalic acid are suggested to have several biological properties, ranging from insecticidal, antifungal, antibiotic, antiviral, herbicidal, hormonal, neurochemical, carcinogenic or antitumoral activities to enzyme and gluconeogenesis inhibitions [15,21]. Studies with the 3T3L1 cell line [25] suggested that CPEFA, such as sterculic and malvalic acids, inhibit Δ9-desaturase activity due to the formation of a non-dissociating enzyme-sterculoyl-CoA complex, preventing the normal desaturation, which occurs in the absence of altered transcription or translation of the Δ9-desaturase gene. In this regard, the well-known inhibitory effect of sterculic acid on Δ9-desaturase activity has been employed as a means to estimate the impact of this enzyme on dairy fat composition, specifically through abomasal infusion of sterculic oil in the bovine [26,27,28] and by jugular infusion of sterculic acid in the caprine and ovine [12,29]. In lactating ewes, the administration of this CPEFA resulted in a 70% inhibition of the enzyme in the mammary gland that persisted partially over time and modified ewe milk fatty acid profile. It resulted in a general increase in the concentration of saturated fatty acids, especially 14:0 and 18:0, and *trans*-11 18:1 (the enzyme substrates) and a decrease in the proportion of *cis*-9-containing fatty acids such as oleic and rumenic acids (the enzyme products). These latter fatty acids are known to have beneficial effects on human health and a crucial role in the regulation of adipocytes proliferation/differentiation, mainly in ruminant species [10,26].

In this context, the determination of sterculic acid and the study of its in vivo occurrence in animals are becoming of notable interest for those biological effects. To the best of our knowledge, no previous publication has reported on CPEFA incorporation in milk fat. Moreover, there is no evidence of its incorporation by the mammary gland.

CPEFA are labile compounds especially in acid media occurring during the esterification process [2,18,22]. As reported above, results of the acid methylation on sterculic acid standard caused alterations, including isomerization. Besides, as suggested by Christie and Han [22], cyclopropenoid together with epoxyl groups in fatty acids are disrupted by acidic conditions and lipid samples containing those compounds should preferably be transesterified with basic reagents. Free fatty acids could be methylated with diazomethane in methanol but both diazomethane and the resulting intermediates are highly toxic and carcinogenic. Taking all these into consideration, a silanyzation process was also carried out on the sterculic acid standard to avoid the acidic conditions. Our results would suggest that silanyzation could be used as a potential alternative derivatization procedure when CPEFA are present in the form of free fatty acids, whereas base-esterification would be recommended when CPEFA are present in the triglyceride form.

In the current research, the presence of sterculic acid was determined in milk triglycerides by GC-MS after base derivatization. The results obtained indicate that the procedure applied was appropriate to prepare sterculic acid methyl ester from milk fat for its determination by GC-MS analysis. In addition, for the first time, our results demonstrated that this CPEFA can be incorporated from the bloodstream into milk fat. 

Sterculic acid concentration reached its maximum concentration after 5 days of systemic administration (Figure 5), reaching mean values of 660 ± 190 mg/kg of milk fat for all treated animals. Despite this concentration being numerically greater on day 5 than on day 4 of administration, the high between-animal variation could rule out differences between both days. This would be further supported by results from the previous study [12], which showed steady state levels of Δ9-desaturase substrates and products on those sampling days. Finally, as expected, CPEFA concentration decreased after stopping its administration and sterculic acid was undetectable in milk fat after 5 days. The mean transfer rate of sterculic acid from blood to milk (8.0 ± 1.0%) was lower than that reported under similar experimental conditions for ^13^C-labelled free fatty acids in sheep from the same dairy breed (15–17% of transference within 72–96 h post-administration via jugular infusion) [30,31]. This low transference suggests that most of the sterculic acid was incorporated or metabolized by other body tissues. Further research would be advisable to examine if this is associated to specificity of acyltransferases in body tissues. In addition, the accumulation of this CPEFA or its metabolites in edible tissues from ruminants (e.g., adipose tissue or liver) needs to be examined, with special attention to implications due to their intake by consumers and considering the acid environment occurring during gastrointestinal digestion.

Overall, our results would represent a valuable starting point supporting the importance of investigating the metabolism of sterculic acid in vivo and encouraging further research to determine CPEFA in dairy fat and its dietary intake from different food/feed sources. 

Research on this issue would be particularly interesting in production systems that involve the use of CPEFA sources in livestock diet (e.g., from cottonseeds). 

## 5. Conclusions

This study demonstrates that a relatively small proportion of sterculic acid is incorporated from the bloodstream of dairy ruminants into milk fat. However, for the first time, a GC-MS methodology to properly detect this CPEFA in animal fat is presented. Our results show that base esterification is appropriate for the analysis of sterculic acid as triglyceride in milk fat, whereas a derivatization based on silanyzation is required for free fatty acid sterculic acid, due to the instability of the cyclopropene ring in acid environment. The systemic administration of 0.5 g sterculic acid/day to mid-lactation ewes by jugular infusion resulted in a maximum concentration of sterculic acid in milk at 660 ± 190 mg/total fat (on average for all treated animals) and, when this administration stopped, sterculic acid quickly decreased and was not detected after 5 days post-treatment. The mean transfer rate of this CPEFA from blood to milk was 8.0 ± 1.0% of the daily dose.

Overall, our research provides a simple methodology for sterculic acid quantification and demonstrates that this CPEFA can be transferred into milk fat. Further studies should focus on determining which animal feed sources would increase sterculic acid levels, especially in milk and other ruminant-derived foods.

## Figures and Tables

**Figure 1 foods-09-00901-f001:**
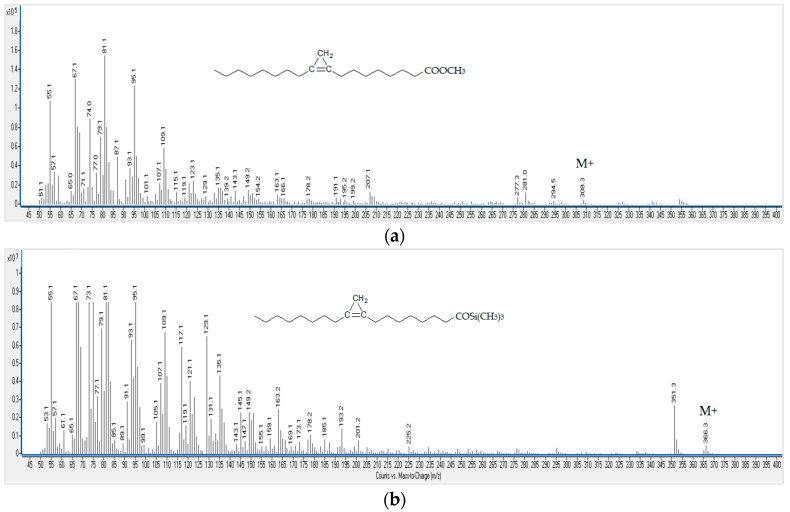
(**a**) Mass spectrum and chemical formula of sterculic acid methyl ester (M^+^: m/z 308); (**b**) Mass spectrum and chemical formula of sterculic acid silyl ether (M^+^: m/z 366).

**Figure 2 foods-09-00901-f002:**
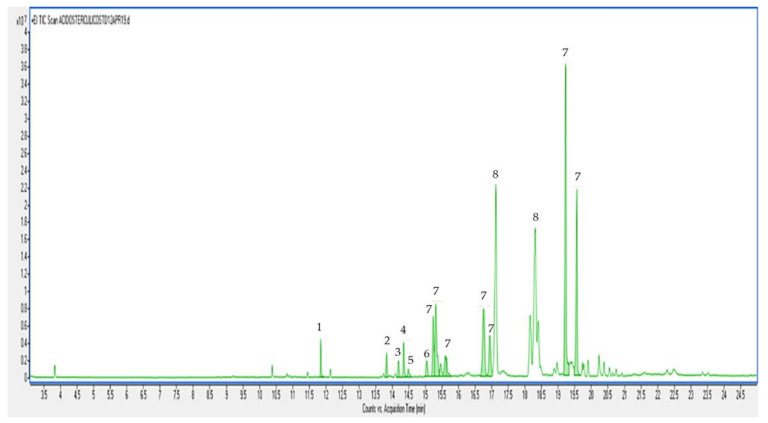
GC-MS chromatogram of methyl esters obtained from the analysis of chemically synthesized sterculic acid after acid esterification. Peaks: (1) palmitic acid (16:0); (2) oleic acid (*cis*-9 18:1); (3) stearic acid (18:0); (4) conjugated linoleic acid (18:2); (5) nonadecanoic acid (19:0); (6) CPFA* (m/z 278; M^+^: m/z 310); (7) CPEFA* (m/z 81; M^+^: m/z 308); (8) not identified* (m/z 85; M^+^: m/z 308). *Mass spectrum was compared with NIST 11 mass spectra libraries and mass spectra published for methyl sterculate and methyl dihydrosterculate [2,17].

**Figure 3 foods-09-00901-f003:**
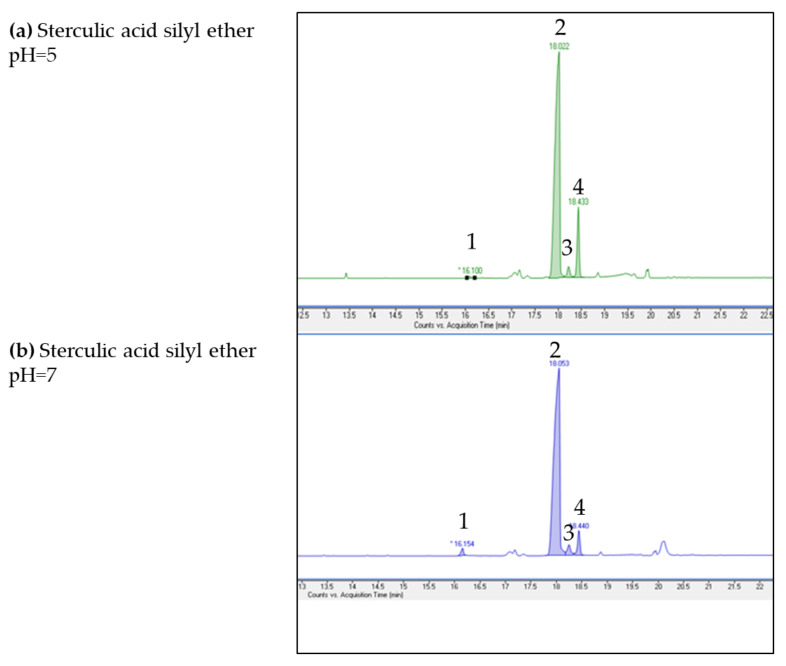
Expanded view of the GC-MS chromatograms obtained from the chemically synthesized sterculic acid standard after silanyzation at (**a**) pH 5 and (**b**) pH 7. Peaks: (1) sterculene (m/z 81; M^+^: m/z 266); (2) sterculic acid (M^+^: m/z 366); (3) dihydrosterculic acid (M^+^: m/z 368); (4) sterculic acid isomer (M^+^: m/z 366).

**Figure 4 foods-09-00901-f004:**
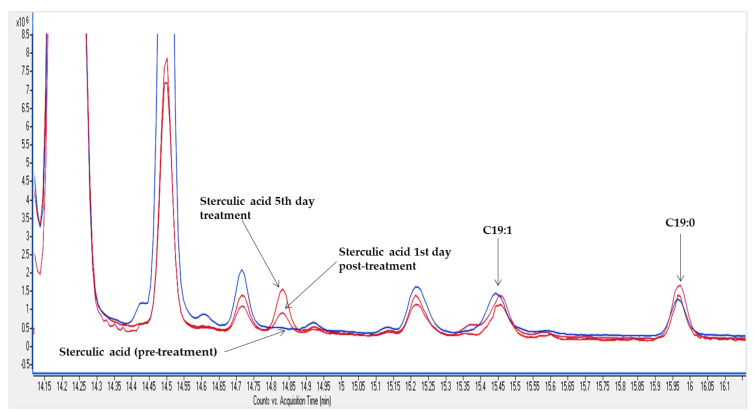
Expanded view of overlaid chromatograms showing the elution zone of sterculic acid, C19:1 and C19:0 methyl esters obtained by GC-MS analysis of ewe milk fat according to the method described by Caligiani et al. [17]. Pre-treatment fat sample (blue) resulted negative to sterculic acid (no signal), whereas sterculic acid was detected at 5th day of treatment (red) and 1st day post-treatment (purple).

**Figure 5 foods-09-00901-f005:**
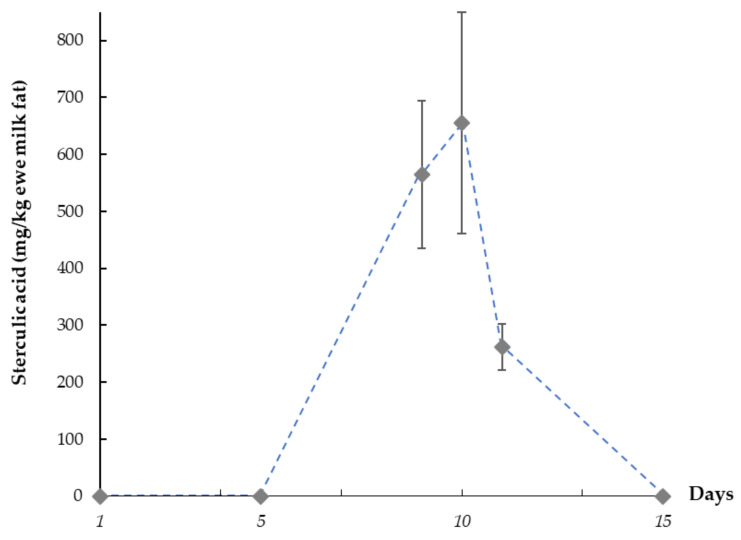
Sterculic acid trend in ewe milk fat over time. *Pre-treatment:* day 1 to 5; *Treatment (0.5 g/day of sterculic acid by jugular infusion):* day 6 to 10; *Post-treatment:* day 11 to 15. Values are reported as mean concentration (mg/kg total fat) of all samples (*n* = 6) at each chosen timepoint: 1st day pre-treatment; 5th day pre-treatment; 4th day-treatment; 5th day-treatment; 1st day-post-treatment; 5th day-post-treatment. Bars indicate standard deviation (as inter-subjects variability).

**Table 1 foods-09-00901-t001:** Molecular ion (M^+^), retention time (RT), and relative percentage (%) of the compounds detected by GC-MS of the chemically synthesized sterculic acid standard after silanyzation at two pH conditions.

			Relative (%) ^1^					
Compound	M^+^ (*m*/*z*)	RT (min)	pH = 5			pH = 7		
Sterculene	266	16.2	12.4	±	17.0	3.6	±	2.7
Sterculic acid	366	18.0	78.4	±	9.4	86.3	±	4.5
Dihydrosterculic acid	368	18.2	2.0	±	1.3	3.7	±	0.4
Sterculic acid isomer	366	18.4	7.2	±	6.3	6.3	±	1.7

^1^ Relative % expressed as mean ± SD of two independent experiments.

**Table 2 foods-09-00901-t002:** Sterculic acid concentration (mg/kg total fat) ^1^ detected in each ewes’ milk fat samples by GC-MS.

Timepoints	Sterculic Acid Concentration (mg/kg Total Fat)					
	Ewe 1	Ewe 2	Ewe 3	Ewe 4	Ewe 5	Ewe 6
1st day-pre-treatment	<LOD ^2^	<LOD	<LOD	<LOD	<LOD	<LOD
5th day-pre-treatment	<LOD	<LOD	<LOD	<LOD	<LOD	<LOD
4th day-treatment	500 ± 40	760 ± 60	394 ± 1	500 ± 20	573 ± 23	654.1 ± 0.3
5th day-treatment	620 ± 50	740 ± 70	555 ± 47	580 ± 40	620 ± 36	825.9 ± 0.1
1st day-post-treatment	210 ± 33	281 ± 48	219 ± 4	296 ± 7	256 ± 5	309 ± 11
5th day-post-treatment	<LOD	<LOD	<LOD	<LOQ ^3^	<LOQ	<LOQ

^1^ Values are the mean ± SD of duplicates from two independent experiments; ^2^ limit of detection (LOD) = 60 mg/kg total fat; ^3^ limit of quantitation (LOQ) = 200 mg/kg total fat.

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
