# Peer review of "Determination of Cyclopropenoid Fatty Acids in Ewe Milk Fat by GC-MS after Intravenous Administration of Sterculic Acid"

_foods, 2020, doi:10.3390/foods9070901_

Round 1

Reviewer 1 Report

This original work presented by Dr. Veronica Lolli and others discussed the quantification of cyclopropenoid fatty acids (CPEFA) in ewe milk following the jugular infusion of sterculic acid. The work is important because the incorporation of CPEFAs in a glandular secretion such as milk has not been previously studied. It outlined the natural sources of CPEFAs and succinctly described their biological relevance as an inhibitor of desaturases. The study is relevant given the potential disruptive effects of CPEFA on lipid synthesis and composition, which could ultimately affect the health of the milk consumer. The paper further described the limitations of conventional GC-MS methods for fatty acid analysis due to the reactivity of CPEFAs.

The authors provided a robust GC-MS methodology for the quantification of CPEFAs. Two derivatization methods were given for sterculic acid: base-catalyzed methyl esterification and silanization. They are complementary as silanization is applicable only to the free fatty acid, while the other is broader in its applicability and can be used to release fatty acids from triglycerides. The authors effectively showed that the sterculic acid in ewe milk was bound in triglycerides. From their results, the authors also calculated the mean transfer rate of sterculic acid from blood to milk. The results and the potential biological effects of CPEFAs were discussed thoroughly.

The following are my comments and suggestions that might improve the manuscript.

  1. The paper would be strengthened by the inclusion of data on calibration, linearity, and LOD and LOQ figures. Some discussion on method reproducibility would also be preferable so that we can better gauge how much of the variation in the data is due to biological variability.
  2. In the discussion (line 286 onwards), the transfer rate of sterculic acid is described to be lower than other free fatty acids. However, from Figure 5 the amount of sterculic acid in milk is still increasing after 5 days of treatment and has not yet reached a steady state. Were the transfer rates of the other fatty acids measured under similar conditions? If both measurements were done before a steady state is achieved, then we might only be seeing a difference in the rate of incorporation. This could have implications on the validity of this comparison.
  3. In the same paragraph, the low transference is ascribed to the incorporation or the metabolism of CPEFA by other body tissues. Could the specificity of acyltransferases also play a part in the incorporation of sterculic acid in milk fats? Have there been studies into this?
  4. The authors mention that CPEFAs are sensitive to acidic conditions. Perhaps the implications of this on the metabolism of ingested CPEFA can be discussed further.

The following are minor points that might improve readability of the manuscript.

  • In line 80, perhaps the authors referring to CPFA, not CPEFA?
  • In the third paragraph of the Discussion (lines 264-265), I would suggest modifying the statement to reflect that no previous publication has reported on CPEFA incorporation in milk.
  • The paper frequently refers to citations as the object of a preposition. See, for example, line 124 (“according to [22]”), lines 139-140 (“described by [23] … reported by [17]”), and line 268 (“as suggested by [22]”). I would suggest referring to the authors instead then providing the citation (for example, “according to Christie et al. [22]”).

Author Response

Response to Reviewer 1 Comments

REVIEWER 1

The following are my comments and suggestions that might improve the manuscript.

  1. The paper would be strengthened by the inclusion of data on calibration, linearity, and LOD and LOQ figures. Some discussion on method reproducibility would also be preferable so that we can better gauge how much of the variation in the data is due to biological variability.

The main objective of the work was to test the presence of sterculic acid in ewe milk fat by GC-MS after its administration at selected timepoints (before, during treatment and after stopping the treatment). So, we especially focused on its detection and identification rather than assessing parameters of the quantitative analysis for validation which were based on previous results on its saturated analogue dihydrosterculic acid in milk fat (for which the method is validated, UNI11650). However, authors realised that LOD and LOQ terms were reported in Table 2 without their values.

Therefore, authors revised the text reporting the reference for the quantitative analysis and specifying the LOD and LOQ (line 226).

Biological variation in sterculic acid amount can be noticed when data are expressed as means of all animals (n=6) (Figure 5) because it does not depend on analytical measurement, whereas it depends on the level of sterculic acid that has been incorporated in milk fat after its administration. To clarify this point text has been little changed (line 237).

  1. In the discussion (line 286 onwards), the transfer rate of sterculic acid is described to be lower than other free fatty acids. However, from Figure 5 the amount of sterculic acid in milk is still increasing after 5 days of treatment and has not yet reached a steady state. Were the transfer rates of the other fatty acids measured under similar conditions? If both measurements were done before a steady state is achieved, then we might only be seeing a difference in the rate of incorporation. This could have implications on the validity of this comparison.

The discussion has been revised to address the reviewer’s comment. Despite milk sterculic acid concentration was numerically greater on day 5 than on day 4 of administration, the high between-animal variation would rule out differences between both days. This would be further supported by results from the previous study, which showed steady state levels of Δ9-desaturase substrates and products on those sampling days. Regarding the comparison between fatty acid transfer rates, we have modified the text to specify that similar experimental conditions were followed in the present and previous studies (fatty acids were always administered intravenously and transfer rates were measured after comparable periods of time).

  1. In the same paragraph, the low transference is ascribed to the incorporation or the metabolism of CPEFA by other body tissues. Could the specificity of acyltransferases also play a part in the incorporation of sterculic acid in milk fats? Have there been studies into this?

Thanks for bringing this to our attention. We completely agree with the reviewer that the presence of specific acyltransferases in different body tissues might help explaining our results but, unfortunately, we are not aware of published studies in this regard. The discussion has been revised to include this point.

  1. The authors mention that CPEFAs are sensitive to acidic conditions. Perhaps the implications of this on the metabolism of ingested CPEFA can be discussed further.

Authors agree with the comment of the reviewer. Author revised the discussion to include this point (see lines 291-295).

The following are minor points that might improve readability of the manuscript.

  • In line 80, perhaps the authors referring to CPFA, not CPEFA?

The sentence in line 80 refers to the analytical conditions adopted to determine CPFA in milk fat were in accordance with analytical conditions adopted by other authors to determine CPEFA in vegetable oils. Authors modified the sentence to avoid misunderstanding.

  • In the third paragraph of the Discussion (lines 264-265), I would suggest modifying the statement to reflect that no previous publication has reported on CPEFA incorporation in milk.

The sentence has been modified according to the suggestion of the reviewer.

  • The paper frequently refers to citations as the object of a preposition. See, for example, line 124 (“according to [22]”), lines 139-140 (“described by [23] … reported by [17]”), and line 268 (“as suggested by [22]”). I would suggest referring to the authors instead then providing the citation (for example, “according to Christie et al. [22]”).

Reference citations have been modified according to the suggestion of the reviewer.

Reviewer 2 Report

I was very impressed with this paper until I came to the discussion, which was not only very limited but also poorly focused on the study described here.  Much of the discussion relates to a preceding paper with these milk samples - discussing the impact of sterculic acid on the overall fatty acid profile in ewes milk, especially the concentrations of fatty acids originating from mammary desaturate activity....However, this paper considers the processes to measure sterculic acid per se in milk - which are very briefly discussed.

The discussion needs to be redrafted around the actual reported results for this paper.  How successful were they at measuring sterculic acid in milk, comparing them with other studies, speculating why they might differ.  Alternative or additional approach might be to discuss the implication of their findings - are there any potetnial impact on consumers or how do the levels infused here with dose rates that might be achieved in reality by foraging livestock

The attached PDF indicates weaknesses that might be addressed

Author Response

Response to Reviewer 2 Comments

REVIEWER 2

R: I was very impressed with this paper until I came to the discussion, which was not only very limited but also poorly focused on the study described here.  Much of the discussion relates to a preceding paper with these milk samples - discussing the impact of sterculic acid on the overall fatty acid profile in ewes milk, especially the concentrations of fatty acids originating from mammary desaturate activity....However, this paper considers the processes to measure sterculic acid per se in milk - which are very briefly discussed. The discussion needs to be redrafted around the actual reported results for this paper.  How successful were they at measuring sterculic acid in milk, comparing them with other studies, speculating why they might differ.  Alternative or additional approach might be to discuss the implication of their findings - are there any potetnial impact on consumers or how do the levels infused here with dose rates that might be achieved in reality by foraging livestock

A: The main objective of the work was to test the presence of sterculic acid in ewe milk fat by GC-MS after its administration at selected timepoints (before, during treatment and after stopping the treatment) for which no information has been previously reported. So, this is the first time that sterculic acid has been detected in animal fat. Our previous information about cyclic fatty acids (especially dihydrosterculic acid) in milk fat were reported to support the methodology adopted. There is a lack of data about the potential effects of sterculic acid on consumer’s health or its occurrence in the livestock. Thus, it is not possible to speculate further about this topic and supports the novelty of our results.

Comments to the attached PDF:

R: Line 49: Is this a paragraph?

A: Text format has been modified.

R: Line 54: Comment if this is commonly referred to as gossypol or another toxic element in cotton seeds

A: Gossypium hirsutum is a specie of Mexican cotton. Authors refer to the seeds belonging to this specie that are an important source of sterculic acid. The presence of gossypol in those seeds or other toxic elements have not been considered for the purposes of the manuscript.

R: Line 179 (Figure 1): the figure title ought to also mention the chemical formula on the charts and explain the M+ possibly leading to Figure 2

A: Authors modified figure caption mentioning the chemical formula and explaining M+ for each derivative.

R: Line 242: if this sentence is necessary at all, it is more appropriate for the introduction rather than the discussion.

A: Author agree with the reviewer. The sentence has been eliminated because it was redundant.

R: Line 246: why controversial?

A: Sterculic acid biological effects are controversial because they are suggested to have beneficial effects such as antifungal, antibiotic, antiviral, antitumoral activities (thus, positive effects for plants, animals and humans)[1] but, on the other hand,  they are suggested to cause some disorders and deleterious effects on animal performance and on animal health[2] (reproductive disorders, growth retardation and altered lipid metabolism).

To avoid misunderstanding, author deleted “controversial” in that sentence.

R: Line 247: does single reference cover these functions?

A: Authors added reference citations.

R: Line 249: Repetition from the introduction and is this relevant to a discussion in a paper measuring their concentrations in milk?

A: Author think that the information reported is not a repetition of the introduction. This paper focused not only on the measurement of sterculic acid in milk fat from an analytical point of view but also on its occurrence in vivo (especially the determination of its chemical form after its incorporation by the mammary gland). So, information has been reported to contextualise the discussion of the results obtained.

R: Line 249: Most of this discussion is relevant to a paper considering the impact of sterculic acid on milk fatty acid profiles- not measuring the presence of sterculic acid itself in the milk-as we have here.

A: Author decided to focus their discussion on the impacts of the determination of sterculic acid in milk fat in the context of milk fatty acids profile quality and to elucidate its in vivo occurrence in animals.

R: Line 285: no record that it was measured in blood? This ought to state from infusion into milk. Speculate why metabolism might differ from that reported in these papers recording 15-17% passing into milk.

A: As described in methods (line 108), sterculic acid has been administrated by jugular infusion (0.5 g/day), so directly in the blood stream. Calculation of the mean transfer rate from blood to milk has been done considering the administered dose and the amount of sterculic acid determined in mik fat.

The same methodology was applied in the other cited papers, thus, facilitating comparison between results. This has been specified in the revised manuscript (from line 287), which has also been modified to further speculate on the possible explanation of differences relative to previous studies, which might be due to specificity of acyltransferases in body tissues.

R: Line 297: add “a relatively small proportion”.

A: the sentence has been modified according to the suggestion of the reviewer.

[1]Peláez, R.; Pariente, A.; Pérez-Sala, Á.; Larráyoz, I.M. Sterculic Acid: The Mechanisms of Action beyond Stearoyl-CoA Desaturase Inhibition and Therapeutic Opportunities in Human Diseases. Cells 2020, 9, 140.

[2] Heuzé, V.; Tran, G.; Hassoun, P.; Bastianelli, D.; Lebas, F. Cottonseed meal. Feedipedia, a programme by INRA, CIRAD, AFZ and FAO. Available online: https://feedipedia.org/node/550. Last updated on 8th February 2019. Accessed on 27th March 2020.

Yu, X.H.; Rawat, R.; Shanklin, J. Characterization and analysis of the cotton cyclopropane fatty acid synthase family and their contribution to cyclopropane fatty acid synthesis. BMC Plant Biol. 2011, 11, 97.

Reviewer 3 Report

The article is well written and thus meets the scientific standard. The authors have a good command of the English Language. However, the following comments do not in any way undermine the study. 

  1. What contributed to the larger error bars in Figure 5?
  2.  Why the results were not subjected to statistical evaluation? 
  3.  More current references between 2014 and 2020 could have been provided. 

Author Response

REVIEWER 3

The article is well written and thus meets the scientific standard. The authors have a good command of the English Language. However, the following comments do not in any way undermine the study. 

1. What contributed to the larger error bars in Figure 5?

As reported in figure caption, values in figure 5 are reported as mean concentrations (mg/kg total fat) of sterculic acid for all samples (n=6) at each selected timepoint and SD represents the inter-subjects variability (so, not analytical error).

2. Why the results were not subjected to statistical evaluation?

The aim of the work is the analysis of the presence of sterculic acid in ewe milk fat by GC-MS after its administration at selected timepoints (before, during treatment and after stopping the treatment). Analysis of each fat samples were conducted in duplicate and results on sterculic acid methyl ester concentration were expressed as mean ± SD for repeatability of the GC-MS analysis (Table 2). Then, the average of sterculic acid for all animals at each timepoints was reported to highlight its trend at those timpepoints (Figure 5). Authors think that further statistical analysis is not required for the purpose of the study.

3. More current references between 2014 and 2020 could have been provided

The cited references are the most current references about cyclic fatty acids due to the lack of studies about this topic.